Evidence synthesis  

health and disease and epidemiology

COVID-19, SARS-CoV-2, exit strategy, mathematical modelling, epidemic control, uncertainty

**Author for correspondence:**
Robin N. Thompson
e-mail: robin.thompson@chch.ox.ac.uk

# Key questions for modelling COVID-19 exit strategies

Robin N. Thompson[1,2,3], T. Déirdre Hollingsworth[4], Valerie Isham[5], Daniel Arribas-Bel[6,7], Ben Ashby[8], Tom Britton[9], Peter Challenor[10], Lauren H. K. Chappell[11], Hannah Clapham[12], Nik J. Cunniffe[13], A. Philip Dawid[14], Christl A. Donnelly[15,16], Rosalind M. Eggo[3], Sebastian Funk[3], Nigel Gilbert[17], Paul Glendinning[18], Julia R. Gog[19], William S. Hart[1], Hans Heesterbeek[20], Thomas House[21,22], Matt Keeling[23], István Z. Kiss[24], Mirjam E. Kretzschmar[25], Alun L. Lloyd[26], Emma S. McBryde[27], James M. McCaw[28], Trevelyan J. McKinley[29], Joel C. Miller[30], Martina Morris[31], Philip D. O'Neill[32], Kris V. Parag[16], Carl A. B. Pearson[3,33], Lorenzo Pellis[19], Juliet R. C. Pulliam[33], Joshua V. Ross[34], Gianpaolo Scalia Tomba[35], Bernard W. Silverman[15,36], Claudio J. Struchiner[37], Michael J. Tildesley[23], Pieter Trapman[9], Cerian R. Webb[13], Denis Mollison[38] and Olivier Restif[39]

[1]Mathematical Institute, University of Oxford, Woodstock Road, Oxford OX2 6GG, UK
[2]Christ Church, University of Oxford, St Aldates, Oxford OX1 1DP, UK
[3]Department of Infectious Disease Epidemiology, London School of Hygiene and Tropical Medicine, Keppel Street, London WC1E 7HT, UK
[4]Big Data Institute, University of Oxford, Old Road Campus, Oxford OX3 7LF, UK
[5]Department of Statistical Science, University College London, Gower Street, London WC1E 6BT, UK
[6]School of Environmental Sciences, University of Liverpool, Brownlow Street, Liverpool L3 5DA, UK
[7]The Alan Turing Institute, British Library, 96 Euston Road, London NW1 2DB, UK
[8]Department of Mathematical Sciences, University of Bath, North Road, Bath BA2 7AY, UK
[9]Department of Mathematics, Stockholm University, Kräftriket, 106 91 Stockholm, Sweden
[10]College of Engineering, Mathematical and Physical Sciences, University of Exeter, Exeter EX4 4QE, UK
[11]Department of Plant Sciences, University of Oxford, South Parks Road, Oxford OX1 3RB, UK
[12]Saw Swee Hock School of Public Health, National University of Singapore, 12 Science Drive, Singapore 117549, Singapore
[13]Department of Plant Sciences, University of Cambridge, Downing Street, Cambridge CB2 3EA, UK
[14]Statistical Laboratory, University of Cambridge, Wilberforce Road, Cambridge CB3 0WB, UK
[15]Department of Statistics, University of Oxford, St Giles', Oxford OX1 3LB, UK
[16]MRC Centre for Global Infectious Disease Analysis, Department of Infectious Disease Epidemiology, Imperial College London, Norfolk Place, London W2 1PG, UK
[17]Department of Sociology, University of Surrey, Stag Hill, Guildford GU2 7XH, UK
[18]Department of Mathematics, University of Manchester, Oxford Road, Manchester M13 9PL, UK
[19]Centre for Mathematical Sciences, University of Cambridge, Wilberforce Road, Cambridge CB3 0WA, UK
[20]Department of Population Health Sciences, Utrecht University, Yalelaan, 3584 CL Utrecht, The Netherlands
[21]IBM Research, The Hartree Centre, Daresbury, Warrington WA4 4AD, UK
[22]Mathematics Institute, and [23]Zeeman Institute for Systems Biology and Infectious Disease Epidemiology Research, School of Life Sciences and Mathematics Institute, University of Warwick, Gibbet Hill Road, Coventry CV4 7AL, UK
[24]School of Mathematical and Physical Sciences, University of Sussex, Falmer, Brighton BN1 9QH, UK
[25]Julius Center for Health Sciences and Primary Care, University Medical Center Utrecht, Utrecht University, Heidelberglaan 100, 3584CX Utrecht, The Netherlands
[26]Biomathematics Graduate Program and Department of Mathematics, North Carolina State University, Raleigh, NC 27695, USA
[27]Australian Institute of Tropical Health and Medicine, James Cook University, Townsville, Queensland 4811, Australia
[28]School of Mathematics and Statistics, University of Melbourne, Carlton, Victoria 3010, Australia
[29]College of Medicine and Health, University of Exeter, Barrack Road, Exeter EX2 5DW, UK
[30]Department of Mathematics and Statistics, La Trobe University, Bundoora, Victoria 3086, Australia

[31]Department of Sociology, University of Washington, Savery Hall, Seattle, WA 98195, USA

[32]School of Mathematical Sciences, University of Nottingham, University Park, Nottingham NG7 2RD, UK

[33]South African DSI-NRF Centre of Excellence in Epidemiological Modelling and Analysis (SACEMA), Stellenbosch University, Jonkershoek Road, Stellenbosch 7600, South Africa

[34]School of Mathematical Sciences, University of Adelaide, South Australia 5005, Australia

[35]Department of Mathematics, University of Rome Tor Vergata, 00133 Rome, Italy

[36]Rights Lab, University of Nottingham, Highfield House, Nottingham NG7 2RD, UK

[37]Escola de Matemática Aplicada, Fundação Getúlio Vargas, Praia de Botafogo, 190 Rio de Janeiro, Brazil

[38]Department of Actuarial Mathematics and Statistics, Heriot-Watt University, Edinburgh EH14 4AS, UK

[39]Department of Veterinary Medicine, University of Cambridge, Madingley Road, Cambridge CB3 0ES, UK

RNT, 0000-0001-8545-5212; TDH, 0000-0001-5962-4238;
BA, 0000-0001-5588-7081; PC, 0000-0001-8661-2718;
NJC, 0000-0002-3533-8672; APD, 0000-0002-7410-6882;
RME, 0000-0002-0362-6717; SF, 0000-0002-2842-3406;
NG, 0000-0002-5937-2410; WSH, 0000-0002-2504-6860;
TH, 0000-0001-5835-8062; MK, 0000-0003-4639-4765;
IZK, 0000-0003-1473-6644; ALL, 0000-0002-6389-6321;
JMM, 0000-0002-2452-3098; CABP, 0000-0003-0701-7860;
JVR, 0000-0002-9918-8167; BWS, 0000-0002-4059-2376;
PT, 0000-0003-0569-1659; OR, 0000-0001-9158-853X

Combinations of intense non-pharmaceutical interventions (lockdowns) were introduced worldwide to reduce SARS-CoV-2 transmission. Many governments have begun to implement exit strategies that relax restrictions while attempting to control the risk of a surge in cases. Mathematical modelling has played a central role in guiding interventions, but the challenge of designing optimal exit strategies in the face of ongoing transmission is unprecedented. Here, we report discussions from the Isaac Newton Institute 'Models for an exit strategy' workshop (11–15 May 2020). A diverse community of modellers who are providing evidence to governments worldwide were asked to identify the main questions that, if answered, would allow for more accurate predictions of the effects of different exit strategies. Based on these questions, we propose a roadmap to facilitate the development of reliable models to guide exit strategies. This roadmap requires a global collaborative effort from the scientific community and policymakers, and has three parts: (i) improve estimation of key epidemiological parameters; (ii) understand sources of heterogeneity in populations; and (iii) focus on requirements for data collection, particularly in low-to-middle-income countries. This will provide important information for planning exit strategies that balance socio-economic benefits with public health.

## 1. Introduction

As of 3 August 2020, the coronavirus disease 2019 (COVID-19) pandemic has been responsible for more than 18 million reported cases worldwide, including over 692 000 deaths. Mathematical modelling is playing an important role in guiding interventions to reduce the spread of severe acute respiratory syndrome coronavirus 2 (SARS-CoV-2). Although the impact of the virus has varied significantly across the

world, and different countries have taken different approaches to counter the pandemic, many national governments introduced packages of intense non-pharmaceutical interventions (NPIs), informally known as 'lockdowns'. Although the socio-economic costs (e.g. job losses and long-term mental health effects) are yet to be assessed fully, public health measures have led to substantial reductions in transmission [1–3]. Data from countries such as Sweden and Japan, where epidemic waves peaked without strict lockdowns, will be useful for comparing approaches and conducting retrospective cost–benefit analyses.

As case numbers have either stabilized or declined in many countries, attention has turned to strategies that allow restrictions to be lifted [4,5] in order to alleviate the economic, social and other health costs of lockdowns. However, in countries with active transmission still occurring, daily disease incidence could increase again quickly, while countries that have suppressed community transmission face the risk of transmission reestablishing due to reintroductions. In the absence of a vaccine or sufficient herd immunity to reduce transmission substantially, COVID-19 exit strategies pose unprecedented challenges to policymakers and the scientific community. Given our limited knowledge, and the fact that entire packages of interventions were often introduced in quick succession as case numbers increased, it is challenging to estimate the effects of removing individual measures directly and modelling remains of paramount importance.

We report discussions from the 'Models for an exit strategy' workshop (11–15 May 2020) that took place online as part of the Isaac Newton Institute's 'Infectious Dynamics of Pandemics' programme. We outline progress to date and open questions in modelling exit strategies that arose during discussions at the workshop. Most participants were working actively on COVID-19 at the time of the workshop, often with the aim of providing evidence to governments, public health authorities and the general public to support the pandemic response. After four months of intense model development and data analysis, the workshop gave participants a chance to take stock and openly share their views of the main challenges they are facing. A range of countries was represented, providing a unique forum to discuss the different epidemic dynamics and policies around the world. Although the main focus was on epidemiological models, the interplay with other disciplines formed an integral part of the discussion. The purpose of this article is twofold: to highlight key knowledge gaps hindering current predictions and projections, and to provide a roadmap for modellers and other scientists towards solutions.

Given that SARS-CoV-2 is a newly discovered virus, the evidence base is changing rapidly. To conduct a systematic review, we asked the large group of researchers at the workshop for their expert opinions on the most important open questions, and relevant literature, that will enable exit strategies to be planned with more precision. By inviting contributions from representatives of different countries and areas of expertise (including social scientists, immunologists, epidemic modellers and others), and discussing the expert views raised at the workshop in detail, we sought to reduce geographical and disciplinary biases. All evidence is summarized here in a policy-neutral manner.

The questions in this article have been grouped as follows. First, we discuss outstanding questions for modelling exit strategies that are related to key epidemiological quantities, such as

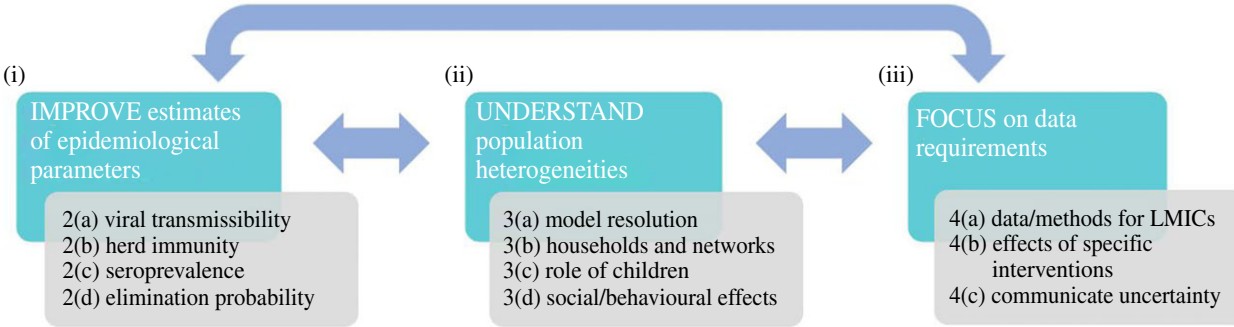

**Figure 1.** Research roadmap to facilitate the development of reliable models to guide exit strategies. Three key steps are required: (i) improve estimates of epidemiological parameters (such as the reproduction number and herd immunity fraction) using data from different countries (§2a–d); (ii) understand heterogeneities within and between populations that affect virus transmission and interventions (§3a–d); and (iii) focus on data requirements for predicting the effects of individual interventions, particularly—but not exclusively—in data-limited settings such as LMICs (§4a–c). Work in these areas must be conducted concurrently; feedback will arise from the results of the proposed research that will be useful for shaping next steps across the different topics. (Online version in colour.)

the reproduction number and herd immunity fraction. We then identify different sources of heterogeneity underlying SARS-CoV-2 transmission and control, and consider how differences between hosts and populations across the world should be included in models. Finally, we discuss current challenges relating to data requirements, focusing on the data that are needed to resolve current knowledge gaps and how uncertainty in modelling outputs can be communicated to policymakers and the wider public. In each case, we outline the most relevant issues, summarize expert knowledge and propose specific steps towards the development of evidence-based exit strategies. This leads to a roadmap for future research (figure 1) made up of three key steps: (i) improve estimation of epidemiological parameters using outbreak data from different countries; (ii) understand heterogeneities within and between populations that affect virus transmission and interventions; and (iii) focus on data needs, particularly data collection and methods for planning exit strategies in low-to-middle-income countries (LMICs) where data are often lacking. This roadmap is not a linear process: improved understanding of each aspect will help to inform other requirements. For example, a clearer understanding of the model resolution required for accurate forecasting (§3a) will inform the data that need to be collected (§4), and vice versa. If this roadmap can be followed, it will be possible to predict the likely effects of different potential exit strategies with increased precision. This is of clear benefit to global health, allowing exit strategies to be chosen that permit interventions to be relaxed while limiting the risk of substantial further transmission.

## 2. Key epidemiological quantities

### (a) How can viral transmissibility be assessed more accurately?

The time-dependent reproduction number, $R(t)$ or $R_t$, has emerged as the main quantity used to assess the transmissibility of SARS-CoV-2 in real time [6–10]. In a population with active virus transmission, the value of $R(t)$ represents the expected number of secondary cases generated by someone infected at time $t$. If this quantity is, and remains below, one, then an ongoing outbreak will eventually fade out.

Although easy to understand intuitively, estimating $R(t)$ from case reports (as opposed to, for example, observing $R(t)$ in known or inferred transmission trees [11]) requires the use of mathematical models. As factors such as contact rates between infectious and susceptible individuals change during an outbreak in response to public health advice or movement restrictions, the value of $R(t)$ has been found to respond rapidly. For example, across the UK, country-wide and regional estimates of $R(t)$ dropped from approximately 2.5–4 in mid-March [7,12] to below one after lockdown was introduced [12,13]. One of the criteria for relaxing the lockdown was for the reproduction number to decrease to 'manageable levels' [14]. Monitoring $R(t)$, as well as case numbers, as individual components of the lockdown are relaxed is critical for understanding whether or not the outbreak remains under control [15].

Several mathematical and statistical methods for estimating temporal changes in the reproduction number have been proposed. Two popular approaches are the Wallinga–Teunis method [16] and the Cori method [17,18]. These methods use case notification data along with an estimate of the serial interval distribution (the times between successive cases in a transmission chain) to infer the value of $R(t)$. Other approaches exist (e.g. based on compartmental epidemiological models [19]), including those that can be used alongside different data (e.g. time series of deaths [7,12,20] or phylogenetic data [21–24]).

Despite this extensive theoretical framework, practical challenges remain. Reproduction number estimates often rely on case notification data that are subject to delays between case onset and being recorded. Available data, therefore, do not include up-to-date knowledge of current numbers of infections, an issue that can be addressed using 'nowcasting' models [8,12,25]. The serial interval represents the period between symptom onset times in a transmission chain, rather than between times at which cases are recorded. Time series of symptom onset dates, or even infection dates (to be used with estimates of the generation interval when inferring $R(t)$), can be estimated from case notification data using latent variable methods [8,26] or methods such as the Richardson–Lucy deconvolution technique [27,28]. The Richardson–Lucy approach has previously been applied to infer incidence curves from time series of deaths [29]. These methods, as well as others that account for reporting delays [30], provide

useful avenues to improve the practical estimation of $R(t)$. Further, changes in testing practice (or capacity to conduct tests) lead to temporal changes in case numbers that cannot be distinguished easily from changes in transmission. Understanding how accurately and how quickly changes in $R(t)$ can be inferred in real time given these challenges is crucial.

Another way to assess temporal changes in $R(t)$, without requiring nowcasting, is by observing people's transmission-relevant behaviour directly, e.g. through contact surveys or mobility data [31]. These methods come with their own limitations: because these surveys do not usually collect data on infections, care must be taken in using them to understand and predict ongoing changes in transmission.

Other outstanding challenges in assessing variations in $R(t)$ include the decrease in accuracy when case numbers are low, and the requirement to account for temporal changes in the serial interval or generation time distribution of the disease [32,33]. When there are few cases (such as in the 'tail' of an epidemic—§2d), there is little information with which to assess virus transmissibility. Methods for estimating $R(t)$ based on the assumption that transmissibility is constant within fixed time periods can be applied with windows of long duration (thereby including more case notification data with which to estimate $R(t)$) [34,35]. However, this comes at the cost of a loss of sensitivity to temporal variations in transmissibility. Consequently, when case numbers are low, the methods described above for tracking transmission-relevant behaviour directly are particularly useful. In those scenarios, the 'transmission potential' might be more important than realized transmission [36].

The effect of population heterogeneity on reproduction number estimates requires further investigation, as current estimates of $R(t)$ tend to be calculated for whole populations (e.g. countries or regions). Understanding the characteristics of constituent groups contributing to this value is important to target interventions effectively [37,38]. For this, data on infections within and between different subpopulations (e.g. infections in care homes and in the wider population) are needed. As well as between subpopulations, it is also necessary to ensure that estimates of $R(t)$ account for heterogeneity in transmission between different infectious hosts. Such heterogeneity alters the effectiveness of different control measures, and, therefore, the predicted disease dynamics when interventions are relaxed. For a range of diseases, a rule of thumb that around 20% of infected individuals are the sources of 80% of infections has been proposed [38,39]. This is supported by recent evidence for COVID-19, which suggests significant individual-level variation in SARS-CoV-2 transmission [40] with some transmission events leading to large numbers of new infections.

Finally, it is well documented that presymptomatic individuals (and, to a lesser extent, asymptomatic infected individuals—i.e. those who never develop symptoms) can transmit SARS-CoV-2 [41,42]. For that reason, negative serial intervals may occur when an infected host displays COVID-19 symptoms before the person who infected them [43,44]. Although methods for estimating $R(t)$ with negative serial intervals exist [44,45], their inclusion in publicly available software for estimating $R(t)$ should be a priority. Increasing the accuracy of estimates of $R(t)$, and supplementing these estimates with other quantities (e.g. estimated epidemic growth rates [46]), is of clear importance. As lockdowns are relaxed, this will permit a fast determination of whether or not removed interventions are leading to a surge in cases.

## (b) What is the herd immunity threshold and when might we reach it?

Herd immunity refers to the accumulation of sufficient immunity in a population through infection and/or vaccination to prevent further substantial outbreaks. It is a major factor in determining exit strategies, but data are still very limited. Dynamically, the threshold at which herd immunity is achieved is the point at which $R(t)$ (§2a) falls below one for an otherwise uncontrolled epidemic, resulting in a negative epidemic growth rate. However, reaching the herd immunity threshold does not mean that the epidemic is over or that there is no risk of further infections. Great care must be taken in communicating this concept to the public, to ensure continued adherence to public health measures. Crucially, whether immunity is gained naturally through infection or through random or targeted vaccination affects the herd immunity threshold, which also depends critically on the immunological characteristics of the pathogen. Since SARS-CoV-2 is a new virus, its immunological characteristics—notably the duration and extent to which prior infection confers protection against future infection, and how these vary across the population—are currently unknown [47]. Lockdown measures have impacted contact structures and hence the accumulation of immunity in the population, and are likely to have led to significant heterogeneity in acquired immunity (e.g. by age, location, workplace). Knowing the extent and distribution of immunity in the population will help guide exit strategies.

As interventions are lifted, whether or not $R(t)$ remains below one depends on the current level of immunity in the population as well as the specific exit strategy followed. A simple illustration is to treat $R(t)$ as a deflation of the original (basic) reproduction number ($R_0$, which is assumed to be greater than one):

$$R(t) = (1 - i(t))(1 - p(t))\,R_0,$$

where $i(t)$ is the immunity level in the community at time $t$ and $p(t)$ is the overall reduction factor from the control measures that are in place. If $i(t) > 1 - 1/R_0$, then $R(t)$ remains below one even when all interventions are lifted: herd immunity is achieved. However, recent results [48,49] show that, for heterogeneous populations, herd immunity occurs at a lower immunity level than $1 - 1/R_0$. The threshold $1 - 1/R_0$ assumes random vaccination, with immunity distributed uniformly in the community. When immunity is obtained from disease exposure, the more socially active individuals in the population are over-represented in cases from the early stages of the epidemic. As a result, the virus preferentially infects individuals with higher numbers of contacts, thereby acting like a well-targeted vaccine. This reduces the herd immunity threshold. However, the extent to which heterogeneity in behaviour lowers the threshold for COVID-19 is currently unknown.

We highlight three key challenges for determining the herd immunity threshold for COVID-19, and hence for understanding the impact of implementing or lifting control measures in different populations. First, most of the quantities for calculating the threshold are not known precisely and require careful investigation. For example, determining the immunity level

in a community is far from trivial for a number of reasons: antibody tests may have variable sensitivity and specificity; it is currently unclear whether or not individuals with mild or no symptoms acquire immunity or test seropositive; the duration of immunity is unknown. Second, estimation of $R_0$, despite receiving significant attention at the start of the pandemic, still needs to be refined within and between countries as issues with early case reports come to light. Third, as discussed in §3, SARS-CoV-2 does not spread uniformly through populations [50]. An improved understanding of the main transmission routes, and which communities are most influential, will help to determine how much lower disease-induced herd immunity is compared to the classical threshold $(1 - 1/R_0)$.

To summarize, it is vital to obtain more accurate estimates of the current immunity levels in different countries and regions, and to understand how population heterogeneity affects transmission and the accumulation of immunity.

## (c) Can seroprevalence surveys provide insight into transmission dynamics and herd immunity?

Quantitative information about current and past infections are key inputs to formulate exit strategies, monitor the progression of epidemics and identify social and demographic sources of transmission heterogeneities. Seroprevalence surveys provide a direct way to estimate the fraction of the population that has been exposed to the virus but has not been detected by regular surveillance mechanisms [51]. Given the possibility of mild or asymptomatic infections, which are not typically included in laboratory-confirmed cases, seroprevalence surveys could be particularly useful for tracking the COVID-19 pandemic [52].

Contacts between pathogens and hosts that elicit an immune response can be revealed by the presence of antibodies. Typically, a rising concentration of immunoglobulin M (IgM) precedes an increase in the concentration of immunoglobulin G (IgG). However, for infections by SARS-CoV-2, there is increasing evidence that IgG and IgM appear concurrently [53]. Most serological assays used for understanding viral transmission measure IgG. Interpretation of a positive result depends on detailed knowledge of immune response dynamics and its epidemiological correspondence to the developmental stage of the pathogen, for example, the presence of virus shedding [54,55]. Serological surveys are common practice in infectious disease epidemiology and have been used to estimate the prevalence of carriers of antibodies, force of infection and reproduction numbers [56], and in certain circumstances (e.g. for measles) to infer population immunity to a pathogen [57]. Unfortunately, a single serological survey only provides information about the number of individuals who are seropositive at the time of the survey (as well as information about the individuals tested, such as their ages [58]). Although information about temporal changes in infections can be obtained by conducting multiple surveys longitudinally [47,59], the precise timings of infections remain unknown.

Available tests vary in sensitivity and specificity, which can impact the accuracy of model predictions if seropositivity is used to assess the proportion of individuals protected from infection or disease. Propagation of uncertainty due to the sensitivity and specificity of the testing procedures and epidemiological interpretation of the immune response are areas that require attention. The possible presence of immunologically silent individuals, as implied by studies of COVID-19 showing that 10–20% of symptomatically infected people have few or no detectable antibodies [60], adds to the known sources of uncertainty.

Many compartmental modelling studies have used data on deaths as the main reliable dataset for model fitting. The extent to which seroprevalence data could provide an additional useful input for model calibration, and help in formulating exit strategies, has yet to be ascertained. With the caveats above, one-off or regular assessments of population seroprevalence could be helpful in understanding SARS-CoV-2 transmission in different locations.

## (d) Is global eradication of SARS-CoV-2 a realistic possibility?

When $R_0$ is greater than one, an emerging outbreak will either grow to infect a substantial proportion of the population or become extinct before it is able to do so [61–65]. If instead $R_0$ is less than one, the outbreak will almost certainly become extinct before a substantial proportion of the population is infected. If new susceptible individuals are introduced into the population (for example, new susceptible individuals are born), it is possible that the disease will persist after its first wave and become endemic [66]. These theoretical results can be extended to populations with household and network structure [67,68] and scenarios in which $R_0$ is very close to one [69].

Epidemiological theory and data from different diseases indicate that extinction can be a slow process, often involving a long 'tail' of cases with significant random fluctuations (electronic supplementary material, figure S1). Long epidemic tails can be driven by spatial heterogeneities, such as differences in weather in different countries (potentially allowing an outbreak to persist by surviving in different locations at different times of year) and varying access to treatment in different locations. Regions or countries that eradicate SARS-CoV-2 successfully might experience reimportations from elsewhere [70,71], for example, the reimportation of the virus to New Zealand from the UK in June 2020.

At the global scale, smallpox is the only previously endemic human disease to have been eradicated, and extinction took many decades of vaccination. The prevalence and incidence of polio and measles have been reduced substantially through vaccination but both diseases persist. The 2001 foot and mouth disease outbreak in the UK and the 2003 SARS pandemic were new epidemics that were driven extinct without vaccination before they became endemic, but both exhibited long tails before eradication was achieved. The 2014–16 Ebola epidemic in West Africa was eliminated (with vaccination at the end of the epidemic [72]), but eradication took some time with flare ups occurring in different countries [73,74].

Past experience, therefore, raises the possibility that SARS-CoV-2 may not be driven to complete extinction in the near future, even if a vaccine is developed and vaccination campaigns are implemented. As exemplified by the Ebola outbreak in the Democratic Republic of the Congo that has only recently been declared over [75], there is an additional challenge of assessing whether the virus really is extinct rather than persisting in individuals who do not report disease [73]. SARS-CoV-2 could become endemic, persisting in populations with limited access to healthcare or circulating in seasonal outbreaks. Appropriate

communication of these scenarios to the public and policymakers—particularly the possibility that SARS-CoV-2 may never be eradicated—is essential.

## 3. Heterogeneities in transmission

### (a) How much resolution is needed when modelling human heterogeneities?

A common challenge faced by epidemic modellers is the tension between making models more complex (and possibly, therefore, seeming more realistic to stakeholders) and maintaining simplicity (for scientific parsimony when data are sparse and for expediency when predictions are required at short notice) [76]. How to strike the correct balance is not a settled question, especially given the increasing amount of available data on human demography and behaviour. Indeed, outputs of multiple models with different levels of complexity can provide useful and complementary information. Many sources of heterogeneity between individuals (and between populations) exist, including the strong skew of severe COVID-19 outcomes towards the elderly and individuals from specific groups. We focus on two sources of heterogeneity in human populations that must be considered when modelling exit strategies: spatial contact structure and health vulnerabilities.

There has been considerable success in modelling local contact structure, both in terms of spatial heterogeneity (distinguishing local and long-distance contacts) and in local mixing structures such as households and workplaces. However, challenges include tracking transmission and assessing changes when contact networks are altered. In spatial models with only a small number of near-neighbour contacts, the number of new infections grows slowly; each generation of infected individuals is only slightly larger than the previous one. As a result, in those models, $R(t)$ cannot significantly exceed its threshold value of one [77]. By contrast, models accounting for transmission within closely interacting groups explicitly contain a mechanism that has a multiplier effect on the value of $R(t)$ [67]. Another challenge is the spatio-temporal structure of human populations: the spatial distribution of individuals is important, but long-distance contacts make populations more connected than in simple percolation-type spatial models [77]. Clustering and pair approximation models can capture some aspects of spatial heterogeneities [78], which can result in exponential rather than linear growth in case numbers [79].

While models can include almost any kind of spatial stratification, ensuring that model outputs are meaningful for exit strategy planning relies on calibration with data. This brings in challenges of merging multiple data types with different stratification levels. For example, case notification data may be aggregated at a regional level within a country, while mobility data from past surveys might be available at finer scales within regions. Another challenge is to determine the appropriate scale at which to introduce or lift interventions. Although measures are usually directed at whole populations within relevant administrative units (country-wide or smaller), more effective interventions and exit strategies may target specific parts of the population [80]. Here, modelling can be helpful to account for

operational costs and imperfect implementation that will offset expected epidemiological gains.

The structure of host vulnerability to disease is generally reported via risk factors, including age, sex and ethnicity [81,82]. From a modelling perspective, a number of open questions exist. To what extent does heterogeneous vulnerability at an individual level affect the impact of exit strategies beyond the reporting of potential outcomes? Where host vulnerability is an issue, is it necessary to account for considerations other than reported risk factors, as these may be proxies for underlying causes? Once communicated to the public, modelling results could create behavioural feedback that might help or hinder exit strategies; some sensitivity analyses would be useful. As with the questions around spatial heterogeneity, modelling variations in host vulnerability could improve proposed exit strategies, and modelling can be used to explore how these are targeted and communicated [5]. Finally, heterogeneities in space and vulnerabilities may interact; modelling these may reveal surprises that can be explored further.

### (b) What are the roles of networks and households in SARS-CoV-2 transmission?

NPIs reduce the opportunity for transmission by breaking up contact networks (closing workplaces and schools, preventing large gatherings), reducing the chance of transmission where links cannot be broken (wearing masks, sneeze barriers) and identifying infected individuals (temperature checks [83], diagnostic testing [84]). Network models [85,86] aim to split pathogen transmission into opportunity (number of contacts) and transmission probability, using data that can be measured directly (through devices such as mobility tracking and contact diaries) and indirectly (through traffic flow and co-occurrence studies). This brings new issues: for example, are observed networks missing key transmission routes, such as indirect contact via contaminated surfaces, or including contacts that are low risk [87]? How we measure and interpret contact networks depends on the geographical and social scales of interest (e.g. wider community spread or closed populations such as prisons and care homes; or sub-populations such as workplaces and schools) and the timescales over which the networks are used to understand or predict transmission.

In reality, individuals belong to households, children attend schools and adults mix in workplaces as well as in social contexts. This has led to the development of household models [67,88–91], multilayer networks [92], bipartite networks [93,94] and networks that are geographically and socially embedded to reflect location and travel habits [95]. These tools can play a key role in understanding and monitoring transmission, and exploring scenarios, at the point of exiting a lockdown: in particular, they can inform whether or not, and how quickly, households or local networks merge to form larger and possibly denser contact networks in which local outbreaks can emerge. Regional variations and socio-economic factors can also be explored.

Contact tracing, followed by isolation or treatment of infected contacts, is a well-established method of disease control. The structure of the contact network is important in determining whether or not contact tracing will be successful. For example, contact tracing in clustered networks is known to be most effective [96,97], since an infected contact can be

traced from multiple different sources. Knowledge of the contact network enhances understanding of the correlation structure that emerges as a result of the epidemic. The first wave of an epidemic will typically infect many of the highly connected nodes and will move slowly to less connected parts of the network, leaving behind islands of susceptible and recovered individuals. This can lead to a correlated structure of susceptible and recovered nodes that may make the networks less vulnerable to later epidemic waves [98], and has implications for herd immunity (§2b).

In heterogeneous populations, relatively few very well-connected people can be major hubs for transmission. Such individuals are often referred to as super-spreaders [99,100] and some theoretical approaches to controlling epidemics are based on targeting them [101]. However, particularly for respiratory diseases, whether specific individuals can be classified as potential super-spreaders, or instead whether any infected individual has the potential to generate super-spreading events, is debated [38,102,103].

As control policies are gradually lifted, the disrupted contact network will start to form again. Understanding how proxies for social networks (which can be measured in near real time using mobility data, electronic sensors or trackers) relate to transmission requires careful consideration. Using observed contacts to predict virus spread might be successful if these quantities are heavily correlated, but one aim of NPIs should be at least a partial decoupling of the two, so that society can reopen but transmission remains controlled. Currently, a key empirical and theoretical challenge is to understand how households are connected and how this is affected by school opening (§3c). An important area for further research is to improve our understanding of the role of within-household transmission in the COVID-19 pandemic. In particular, do sustained infection chains within households lead to amplification of infection rates between households despite lockdowns aimed at minimizing between-household transmission?

Even for well-studied household models, development of methods accommodating time-varying parameters such as variable adherence to household-based policies and/or compensatory behaviour would be valuable. It would be useful to compare interventions and de-escalation procedures in different countries to gain insight into: regional variations in contact and transmission networks; the role of different household structures in transmission and the severity of outcomes (accounting for different household sizes and age-structures); the cost-effectiveness of different policies, such as household-based isolation and quarantine in the UK compared to out-of-household quarantine in Australia and Hong Kong. First Few X (FFX) studies [104,105], now adopted in several countries, provide the opportunity not only to improve our understanding of critical epidemiological characteristics (such as incubation periods, generation intervals and the roles of asymptomatic and presymptomatic transmission) but also to make many of these comparisons.

## (c) What is the role of children in SARS-CoV-2 transmission?

A widely implemented early intervention was school closure, which is frequently used during influenza pandemics [106,107]. Further, playgrounds were closed and social distancing has kept children separated. However, the role of children

in SARS-CoV-2 transmission is unclear. Early signs from Wuhan (China), echoed elsewhere, showed many fewer cases in under 20s than expected. There are three aspects of the role of children in transmission: (i) susceptibility; (ii) infectiousness once infected; and (iii) propensity to develop disease if infected [108,109]. Evidence for age-dependent susceptibility and infectiousness is mixed, with infectiousness the more difficult to quantify. However, evidence is emerging of lower susceptibility to infection in children compared to adults [110], although the mechanism underlying this is unknown and it may not be generalizable to all settings. Once infected, children appear to have a milder course of infection, and it has been suggested that children have a higher probability of a fully subclinical course of infection.

Reopening schools is of clear importance both in ensuring equal access to education and enabling carers to return to work. However, the transmission risk within schools and the potential impact on community transmission needs to be understood so that policymakers can balance the potential benefits and harms. As schools begin to reopen, there are major knowledge gaps that prevent clear answers. The most pressing question is the extent to which school restarting will affect population-level transmission, characterized by $R(t)$ (§2a). Clearer quantification of the role of children could have come from analysing the effects of school closures in different countries in February and March, but closures generally coincided with other interventions and so it has proved difficult to unpick the effects of individual measures [7]. Almost all schools in Sweden stayed open to under-16s (with the exception of one school that closed for two weeks [111]), and schools in some other countries are beginning to reopen with social distancing measures in place, providing a potential opportunity to understand within-school transmission more clearly. Models can also inform the design of studies to generate the data required to answer key questions.

The effect of opening schools on $R(t)$ also depends on other changes in the community. Children, teachers and support staff are members of households; lifting restrictions may affect all members. Modelling school reopening must account for all changes in contacts of household members [112], noting that the impact on $R(t)$ may depend on the other interventions in place at that time. The relative risk of restarting different school years (or universities) does not affect the population $R(t)$ straightforwardly, since older children tend to live with adults who are older (compared to younger children), and households with older individuals are at greater risk of severe outcomes. Thus, decisions about which age groups return to school first and how they are grouped at school must balance the risks of transmission between children, transmission to and between their teachers, and transmission to and within the households of the children and teachers.

Return to school affects the number of physical contacts of teachers and support staff. Schools will not be the same environments as prior to lockdown, since physical distancing measures will be in place. These include smaller classes and changes in layout, plus increased hygiene measures. Some children and teachers may be less likely to return to school because of underlying health conditions and if there is transmission within schools, there may be absenteeism following infection. Models must, therefore, consider the different effects on transmission of pre- and post-lockdown school

environments. Post-lockdown, with social distancing in place in the wider community, reopening schools could link subcommunities of the population together, and models can be used to estimate the wider effects on population transmission as well as within schools. These estimates are likely to play a central role in decisions surrounding when and how to reopen schools.

## (d) The pandemic is social: how can we model that?

While the effects of population structure and heterogeneities can be approximated in standard compartmental epidemiological models [2,73,113], such models can become highly complex and cumbersome to specify and solve as more heterogeneities are introduced. An alternative approach is agent-based modelling. Agent-based models (ABM) allow complex systems such as societies to be represented, using virtual agents programmed to have behavioural and individual characteristics (age, sex, ethnicity, income, employment status, etc.) as well as the capacity to interact with other agents [114]. In addition, ABM can include societal-level factors such as the influence of social media, regulations and laws, and community norms. In more sophisticated ABM, agents can anticipate and react to scenarios, and learn by trial and error or by imitation. ABM can represent systems in which there are feedbacks, tipping points, the emergence of higher-level properties from the actions of individual agents, adaptation and multiple scales of organization—all features of the COVID-19 pandemic and societal reactions to it.

While ABM arise from a different tradition, they can incorporate the insights of compartmental models; for example, agents must transition through disease states (or compartments) such that the mean transition rates correspond to those in compartmental models. However, building an ABM that represents a population on a national scale is a huge challenge and is unlikely be accomplished in a timescale useful for the current pandemic. ABM often include many parameters, leading to challenges of model parametrization and a requirement for careful uncertainty quantification and sensitivity analyses to different inputs. On the other hand, useful ABM do not have to be all-encompassing. There are already several models that illustrate the effects of policies such as social distancing on small simulated populations. These models can be very helpful as 'thought experiments' to identify the potential effects of candidate policies such as school re-opening and restrictions on long-distance travel, as well as the consequences of non-compliance with government edicts.

There are two areas where long-term action should be taken. First, more data about people's ordinary behaviour are required: what individuals do each day (through time-use diaries), whom they meet (possibly through mobile phone data, if consent can be obtained) and how they understand and act on government regulation, social media influences and broadcast information [115]. Second, a large, modular ABM should be built that represents heterogeneities in populations and that is properly calibrated as a social 'digital twin' of our own society, with which we can carry out virtual policy experiments. Had these developments occurred before, they would have been useful currently. As a result, if these are addressed now, they will aid the planning of future exit strategies.

## 4. Data needs and communicating uncertainty

### (a) What are the additional challenges of data-limited settings?

In most countries, criteria for ending COVID-19 lockdowns rely on tracking trends in numbers of confirmed cases and deaths, and assessments of transmissibility (§2a). This section focuses on the relaxation of interventions in LMICs, although many issues apply everywhere. Perhaps surprisingly, concerns relating to data availability and reliability (e.g. lack of clarity about sampling frames) remain worldwide. Other difficulties have also been experienced in many countries throughout the pandemic (e.g. shortages of vital supplies, perhaps due in developed countries to previous emphasis on healthcare system efficiency rather than pandemic preparedness [116]).

Data about the COVID-19 pandemic and about the general population and context can be unreliable or lacking globally. However, due to limited healthcare access and utilization, there can be fewer opportunities for diagnosis and subsequent confirmation of cases in LMICs compared to other settings, unless there are active programmes [117]. Distrust can make monitoring programmes difficult, and complicate control activities like test–trace–isolate campaigns [118,119]. Other options for monitoring—such as assessing excess disease from general reporting of acute respiratory infections or influenza-like illness—require historical baselines that may not exist [120,121]. In general, while many LMICs will have a well-served fraction of the population, dense peri-urban and informal settlements are typically outside that population and may rapidly become a primary concern for transmission [122]. Since confirmed case numbers in these populations are unlikely to provide an accurate representation of the underlying epidemic, reliance on alternative data such as clinically diagnosed cases may be necessary to understand the epidemic trajectory. Some tools for rapid assessment of mortality in countries where the numbers of COVID-19-related deaths are hard to track are starting to become available [123].

In settings where additional data collection is not affordable, models may provide a clearer picture by incorporating available metadata, such as testing and reporting rates through time, sample backlogs and suspected COVID-19 cases based on syndromic surveillance. By identifying the most informative data, modelling could encourage countries to share available data more widely. For example, burial reports and death certificates may be available, and these data can provide information on the demographics that influence the infection fatality rate. These can in turn reveal potential COVID-19 deaths classified as other causes and hence missing from COVID-19 attributed death notifications.

In addition to the challenges in understanding the pandemic in these settings, metrics on health system capacity (including resources such as beds and ventilators), as needed to set targets for control, are often poorly documented [124]. Furthermore, the economic hardships and competing health priorities in low-resource settings change the objectives of lifting restrictions—for example, hunger due to loss of jobs and changes in access to routine healthcare (e.g. HIV services and childhood vaccinations) as a result of lockdown have the potential to cost many lives in themselves, both in the short and long term [125,126]. This must be accounted for when deciding how to relax COVID-19 interventions.

We have identified three key challenges for epidemic modellers to help guide exit strategies in data-limited settings: (i) explore policy responses that are robust to missing information; (ii) conduct value-of-information analyses to prioritize additional data collection; and (iii) develop methods that use metadata to interpret epidemiological patterns.

In general, supporting LMICs calls for creativity in the data that are used to parametrize models and in the response activities that are undertaken. Some LMICs have managed the COVID-19 pandemic successfully so far (e.g. Vietnam, as well as Trinidad and Tobago [127]). However, additional support in LMICs is required and warrants special attention. If interventions are relaxed too soon, fragile healthcare systems may be overwhelmed. If instead they are relaxed too late, socio-economic consequences can be particularly severe.

## (b) Which data should be collected as countries emerge from lockdown, and why?

Identifying the effects of the different components of lockdown is important to understand how—and in which order—interventions should be released. The impact of previous measures must be understood both to inform policy in real time and to ensure that lessons can be learnt.

All models require information to make their predictions relevant. Data from PCR tests for the presence of active virus and serological tests for antibodies, together with data on COVID-19-related deaths, are freely available via a number of internet sites (e.g. [128]). However, metadata associated with testing protocols (e.g. reason for testing, type of test, breakdowns by age and underlying health conditions) and the definition of COVID-19-related death, which are needed to quantify sources of potential bias and parametrize models correctly, are often unavailable. Data from individuals likely to have been exposed to the virus (e.g. within households of known infected individuals), but who may or may not have contracted it themselves, are also useful for model parametrization [129]. New sources of data range from tracking data from mobile phones [130] to social media surveys [131] and details of interactions with public health providers [132]. Although potentially valuable, these data sources bring with them biases that are not always understood. These types of data are also often subject to data protection and/or costly fees, meaning that they are not readily available to all scientists. Mixing patterns by age were reasonably well-characterized before the current pandemic [133,134] (particularly for adults of different ages) and have been used extensively in existing models. However, there are gaps in these data and uncertainty in the impacts that different interventions have had on mixing. Predictive models for policy tend to make broad assumptions about the effects of elements of social distancing [135], although results of studies that attempt to estimate effects in a more data-driven way are beginning to emerge [136]. The future success of modelling to understand when controls should be relaxed or tightened depends critically on whether, and how accurately as well as how quickly, the effects of different elements of lockdown can be parametrized.

Given the many differences in lockdown implementation between countries, cross-country comparisons offer an opportunity to estimate the effects on transmission of each component of lockdown [7]. However, there are many challenges in comparing SARS-CoV-2 dynamics in different countries. Alongside variability in the timing, type and impact of interventions, the numbers of importations from elsewhere will vary [70,137]. Underlying differences in mixing, behavioural changes in response to the pandemic, household structures, occupations and distributions of ages and co-morbidities are likely to be important but uncertain drivers of transmission patterns. A current research target is to understand the role of weather and climate in SARS-CoV-2 transmission and severity [138]. Many analyses across and within countries highlight potential correlations between environmental variables and transmission [139–144], although sometimes by applying ecological niche modelling frameworks that may be ill-suited for modelling a rapidly spreading pathogen [145–147]. Assessments of the interactions between weather and viral transmissibility are facilitated by the availability of extensive datasets describing weather patterns, such as the European Centre for Medium-Range Weather Forecasts ERA5 dataset [148] and simulations of the Community Earth System Model that can be used to estimate the past, present and future values of meteorological variables worldwide [149]. Temperature, humidity and precipitation are likely to affect the survival of SARS-CoV-2 outside the body, and prevailing weather conditions could, in theory, tip $R(t)$ above or below one. However, the effects of these factors on transmission have not been established conclusively, and the impact of seasonality on short- or long-term SARS-CoV-2 dynamics is likely to depend on other factors including the timing and impact of interventions, and the dynamics of immunity [47,150]. It is hard to separate the effect of the weather on virus survival from other factors including behavioural changes in different seasons [151]. The challenge of disentangling the impact of variations in weather on transmission from other epidemiological drivers in different locations is, therefore, a complex open problem.

In seeking to understand and compare COVID-19 data from different countries, there is a need to coordinate the design of epidemiological studies, involving longitudinal data collection and case–control studies. This will help enable models to track the progress of the epidemic and the impacts of control policies internationally. It will also allow more refined conclusions than those that follow from population data alone. Countries with substantial epidemiological modelling expertise should support epidemiologists elsewhere with standardized protocols for collecting data and using models to inform policy. There is a need to share models to be used 'in the field'. Collectively, these efforts will ensure that models are parametrized as realistically as possible for particular settings. In turn, as interventions are relaxed, this will allow us to detect the earliest possible reliable signatures of a resurgence in cases, leading to an unambiguous characterization of when it is necessary for interventions to be reintroduced.

## (c) How should model and parameter uncertainty be communicated?

SARS-CoV-2 transmission models have played a crucial role in shaping policies in different countries, and their predictions have been a regular feature of media coverage of the pandemic [135,152]. Understandably, both policymakers and journalists generally prefer single 'best guess' figures from models, rather than a range of plausible values. However, the ranges of outputs that modellers provide include important information about the variety of possible scenarios and guard

against over-interpretation of model results. Not displaying information about uncertainty can convey a false confidence in predictions. It is critical that modellers present uncertainty in a way that is understandable and useful for policymakers and the public [76].

There are numerous and often inextricable ways in which uncertainty enters the modelling process. Model assumptions inevitably vary according to judgements regarding which features are included [1,95] and which datasets are used to inform the model [153]. Within any model, ranges of parameter values can be considered to allow for uncertainty about clinical characteristics of COVID-19 (e.g. the infectious period and case fatality rate) [154]. Alternative initial conditions (e.g. numbers and locations of imported cases seeding national outbreaks, or levels of population susceptibility) can be considered. In modelling exit strategies, when surges in cases starting from small numbers may occur and where predictions will depend on characterizing epidemiological parameters as accurately as possible, stochastic models may be of particular importance. Not all the uncertainty arising from such stochasticity will be reduced by collecting more data; it is inherent to the process.

Where models have been developed for similar purposes, formal methods of comparison can be applied, but in epidemiological modelling, models often have been developed to address different questions, possibly involving 'what-if?' scenarios, in which case only qualitative comparisons can be made. The ideal outcome is when different models generate similar conclusions, demonstrating robustness to the detailed assumptions. Where there is a narrowly defined requirement, such as short-term predictions of cases and deaths, more tractable tools for comparing the outputs from different models in real time would be valuable. One possible approach is to assess the models' past predictive performance [33,155]. Ensemble estimates, most commonly applied for forecasting disease trajectories, allow multiple models' predictions to be combined [156,157]. The assessment of past performance can then be used to weight models in the ensemble. Such approaches typically lead to improved point and variance estimates.

To deal with parameter uncertainty, a common approach is to perform sensitivity analyses in which model parameters are repeatedly sampled from a range of plausible values, and the resulting model predictions compared; both classical and Bayesian statistical approaches can be employed [158–160]. Methods of uncertainty quantification provide a framework in which uncertainties in model structure, epidemiological parameters and data can be considered together. In practice, there is usually only a limited number of policies that can be implemented. An important question is often whether or not the optimal policy can be identified given the uncertainties we have described, and decision analyses can be helpful for this [161,162].

In summary, communication of uncertainty to policymakers and the general public is challenging. Different levels of detail may be required for different audiences. There are many subtleties: for instance, almost any epidemic model can provide an acceptable fit to data in the early phase of an outbreak, since most models predict exponential growth. This can induce an artificial belief that the model must be based on sensible underlying assumptions, and the true uncertainty about such assumptions has vanished. Clear presentation of data is critical. It is important not simply to present data on the numbers of cases, but also on the numbers of individuals who have been tested. Clear statements of the individual values used to calculate quantities such as the case fatality rate are vital, so that studies can be interpreted and compared correctly [163,164]. Going forwards, improved communication of uncertainty is essential as models are used to predict the effects of different exit strategies.

## 5. Summary and discussion

We have highlighted ongoing challenges in modelling the COVID-19 pandemic, and uncertainties faced devising lockdown exit strategies. It is important, however, to put these issues into context: at the start of 2020, SARS-CoV-2 was unknown, and its pandemic potential only became apparent at the end of January. The speed with which the scientific and public health communities came together and the openness in sharing data, methods and analyses are unprecedented. At very short notice, epidemic modellers mobilized a substantial workforce—mostly on a voluntary basis—and state-of-the-art computational models. Far from the rough-and-ready tools sometimes depicted in the media, the modelling effort deployed since January is a collective and multi-pronged effort benefitting from years of experience of epidemic modelling, combined with long-term engagement with public health agencies and policymakers.

Drawing on this collective expertise, the virtual workshop convened in mid-May by the Isaac Newton Institute generated a clear overview of the steps needed to improve and validate the scientific advice to guide lockdown exit strategies. Importantly, the roadmap outlined in this paper is meant to be feasible within the lifetime of the pandemic. Infectious disease epidemiology does not have the luxury of waiting for all data to become available before models must be developed. As discussed here, the solution lies in using diverse and flexible modelling frameworks that can be revised and improved iteratively as more data become available. Equally important is the ability to assess the data critically and bring together evidence from multiple fields: numbers of cases and deaths reported by regional or national authorities only represent a single source of data, and expert knowledge is even required to interpret these data correctly.

In this spirit, our first recommendation is to improve estimates of key epidemiological parameters. This requires close collaboration between modellers and the individuals and organizations that collect epidemic data, so that the caveats and assumptions on each side are clearly presented and understood. That is a key message from the first section of this study, in which the relevance of theoretical concepts and model parameters in the real world was demonstrated: far from ignoring the complexity of the pandemic, models draw from different sources of expertise to make sense of imperfect observations. By acknowledging the simplifying assumptions of models, we can assess the models' relative impacts and validate or replace them as new evidence becomes available.

Our second recommendation is to seek to understand important sources of heterogeneity that appear to be driving the pandemic and its response to interventions. Agent-based modelling represents one possible framework for modelling complex dynamics, but standard epidemic models can also be extended to include age groups or any other relevant strata in the population as well as spatial structure. Network

models provide computationally efficient approaches to capture different types of epidemiological and social interactions. Importantly, many modelling frameworks provide avenues for collaboration with other fields, such as the social sciences.

Our third and final recommendation regards the need to focus on data requirements, particularly (although not exclusively) in resource-limited settings such as LMICs. Understanding the data required for accurate predictions in different countries requires close communication between modellers and governments, public health authorities and the general public. While this pandemic casts a light on social inequalities between and within countries, modellers have a crucial role to play in sharing knowledge and expertise with those who need it most. During the pandemic so far, countries that might be considered similar in many respects have often differed in their policies; either in the choice or the timing of restrictions imposed on their respective populations. Models are important for drawing reliable inferences from global comparisons of the relative impacts of different interventions. All too often, national death tolls have been used for political purposes in the media, attributing the apparent success or failure of particular countries to specific policies without presenting any convincing evidence. Modellers must work closely with policymakers, journalists and social scientists to improve the communication of rapidly changing scientific knowledge while conveying the multiple sources of uncertainty in a meaningful way.

We are now moving into a stage of the COVID-19 pandemic in which data collection and novel research to inform the modelling issues discussed here are both possible and essential for global health. These are international challenges that require an international collaborative response from diverse scientific communities, which we hope that this article will stimulate. This is of critical importance, not only to tackle this pandemic but also to improve the response to future epidemics of emerging infectious diseases.

Data accessibility. Data sharing is not applicable to this manuscript as no new data were created or analysed in this study.

Authors' contributions. R.N.T., T.D.H., V.I., H.H., D.M. and O.R. organized the workshop and designed the study. All authors attended the workshop, contributed to discussions and wrote sections of the manuscript. R.N.T. compiled the manuscript. All authors edited the manuscript and approved the final version for publication.

Competing interests. The authors declare that no competing interests exist.

Funding. This work was supported by the Isaac Newton Institute (EPSRC grant no. EP/R014604/1). R.N.T. thanks Christ Church (Oxford) for funding via a Junior Research Fellowship. R.N.T. and S.F. acknowledge support from the Wellcome Trust (grant no. 210758/Z/18/Z). L.H.K.C. acknowledges support from the BBSRC (grant no. BB/R009236/1). B.A. is supported by the Natural Environment Research Council (grant no. NE/N014979/1). C.A.D. and K.V.P. thank the UK MRC and DFID for centre funding (grant no. MR/R015600/1). C.A.D. also thanks the UK NIHR (National Institute for Health Research) HPRU (Health Protection Research Unit). R.M.E. acknowledges HDR UK (grant no. MR/S003975/1) and the UK MRC (grant no. MC_PC 19065). H.H. and M.E.K. acknowledge support from the Netherlands Organization for Health Research and Development (ZonMw; grant no. 10430022010001). T.H. acknowledges support from the Royal Society (grant no. INF\R2\180067) and the Alan Turing Institute for Data Science and Artificial Intelligence. M.K. and M.J.T. acknowledge support from the UK MRC (grant no. MR/V009761/1). I.Z.K. acknowledges support from the Leverhulme Trust (grant no. RPG-2017-370). M.E.K. acknowledges support from the Netherlands Organization for Health Research and Development (ZonMw; grant no. 91216062). J.C.M. acknowledges start-up funding from La Trobe University. C.A.B.P. acknowledges funding of the NTD Modelling Consortium by the Bill and Melinda Gates Foundation (grant no. OPP1184344). L.P. acknowledges support from the Wellcome Trust and the Royal Society (grant no. 202562/Z/16/Z). J.R.C.P. acknowledges support from the South African Centre for Epidemiological Modelling and Analysis (SACEMA), a Department of Science and Innovation-National Research Foundation Centre of Excellence hosted at Stellenbosch University. C.J.S. acknowledges support from CNPq and FAPERJ. P.T. acknowledges support from Vetenskapsrådet Swedish Research Council (grant no. 2016-04566).

Acknowledgements. Thanks to the Isaac Newton Institute for Mathematical Sciences, Cambridge (www.newton.ac.uk), for support during the virtual 'Infectious Dynamics of Pandemics' programme. This work was undertaken in part as a contribution to the 'Rapid Assistance in Modelling the Pandemic' initiative coordinated by the Royal Society. Thanks to Sam Abbott for helpful comments about the manuscript.

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
