## [Reviewer comments · Proceedings of the Royal Society B: Biological Sciences]

Review History

RSPB-2020-1405.R0 (Original submission)

Review form: Reviewer 1

Recommendation

Accept with minor revision (please list in comments)

Scientific importance: Is the manuscript an original and important contribution to its field?

Excellent

General interest: Is the paper of sufficient general interest?

Excellent

Quality of the paper: Is the overall quality of the paper suitable?

Excellent

Is the length of the paper justified?

Yes

Should the paper be seen by a specialist statistical reviewer?

No

Do you have any concerns about statistical analyses in this paper? If so, please specify them explicitly in your report.

No

It is a condition of publication that authors make their supporting data, code and materials available - either as supplementary material or hosted in an external repository. Please rate, if applicable, the supporting data on the following criteria.

Is it accessible?

N/A

Is it clear?

N/A

Is it adequate?

N/A

Do you have any ethical concerns with this paper?

No

Comments to the Author

This is an excellent review of the key issues regarding COVID-19 epidemic modelling at this stage of the pandemic. The paper focusses on the important and timely question of exit from lockdown. But I believe that it has wider relevance than that and will become a 'go to' reference concerning the use of epidemiological models during a public health emergency. None of the issues raised are new, but there is a need for re-stating the problems and solutions, if only because they are being re-visited during the current crisis.

I have a number of suggestions - all minor - that the authors might wish to consider.

L106. Personally, I'd delete the word "strong". There are several anomalies, not least the peaking of the epidemics in Sweden and Japan without strict 'lockdowns' (though social distancing was still surely a key factor). Ref. [1] reports an analysis that (to put it as diplomatically as possible) will surely not be the last word on the impact of lockdowns.

L157. I like the road map and it is highlighted again in the conclusions. But is the process really linear? I'd suggest there are feedbacks at every stage and particularly from the gathering of data to refinement of model structure and parameterisation. A good example for COVID-19 would be the need to include care homes in any detailed model. That need emerged (sadly, rather belatedly) from observation; the early models missed it entirely.

L173. I'd insert "purely" here. In other epidemics (FMD 2001 being a good example) it was possible to estimate $R(t)$ directly from contact tracing data.

LL184-7. Here the concept of "instantaneous" R is raised. It's never fully explained and never mentioned again so seems a distraction.

L227. Interesting point about negative serial intervals. I'm unsure whether this actually occurs (indeed, I would be suspicious of the data if it did) but it does suggest a related complication:

transmission from cases that are asymptomatic throughout.

LL242-3. Parentheses not required. That condition is not an add-on, it's crucial.

LL313-6. In general, the manuscript is extremely well written. But after several attempts I still couldn't make sense of this sentence.

Section 1.4. This section makes an important but slightly unsatisfying point about epidemic tails. The empirical evidence for COVID-19 epidemic tails is already plain to see. But the manuscript is less explicit about what models say about mechanisms. Presumably one answer is spatial (and presumably other) kinds of heterogeneity. Are there others? Also, what happens to $R(t)$ during a tail? Empirically it will rise to be close to 1, but that may not reflect any change in the underlying dynamics - a looming point of confusion as case numbers fall in the UK and elsewhere.

LL542-544. The ordering of these types of transmission suggests the wrong prioritisation. There is good evidence now from the world and from the UK, that teacher-teacher and teacher-child transmissions are the most important in a school. Child-teacher transmissions are extremely rare: none from any published study so far that I am aware of, and just one instance from PHE data on school "outbreaks".

Section 3, Data Needs. This section jars slightly because of an apparent shifting of attention to LMICs, which are not emphasized in previous sections. I noted this because the nature and quality of the epidemic data available in the UK is a huge concern, even now. Lack of clarity about sampling frame, sometimes even lack of denominator data, has made epidemiological analysis more difficult.

I suppose ref. 98 is fairly described as a starting point but it is too incomplete and error-strewn to be used uncritically in any formal analysis.

Review form: Reviewer 2

Recommendation

Major revision is needed (please make suggestions in comments)

Scientific importance: Is the manuscript an original and important contribution to its field?

Good

General interest: Is the paper of sufficient general interest?

Good

Quality of the paper: Is the overall quality of the paper suitable?

Good

Is the length of the paper justified?

Yes

Should the paper be seen by a specialist statistical reviewer?

No

Do you have any concerns about statistical analyses in this paper? If so, please specify them explicitly in your report.

No

It is a condition of publication that authors make their supporting data, code and materials available - either as supplementary material or hosted in an external repository. Please rate, if applicable, the supporting data on the following criteria.

Is it accessible?

N/A

Is it clear?

N/A

Is it adequate?

N/A

Do you have any ethical concerns with this paper?

No

Comments to the Author

This is an interesting and insightful discussion of the complexities of modelling COVID-19 exit strategies, with a global perspective.

One aspect that does not appear to have been considered is the role of weather conditions in COVID transmission as part of developing modelling - while there seems to be consensus that COVID-19 transmission is not likely to be halted by any particular set of weather conditions, there are a number of published studies suggesting that factors such as temperature, humidity, precipitation, solar radiation, etc. have effects of significant sizes on transmissibility. It is hard to separate the direct effects of weather on the virus in outdoor environments from the modulating effect of weather on human behaviour in relation to compliance with lockdown conditions and balance of time spent in indoor and outdoor environments. A number of studies have noted that weather conditions in the major foci of Wuhan, Daegu, Qom and Lombardy in January/February 2020 were closely similar. It is reasonable to consider that prevailing weather conditions in a particular place and time might be able to tip $R(t)$ to one side or the other of one, and thus be the basis for tweaking other NPI components accordingly. One good thing is that global gridded weather data, for example the ERA5-T model, are available in high resolution globally and are in the public domain.

Some minor points:

Line 362 – my understanding is that the long-running Ebola outbreak in eastern DR Congo has now been declared over, although there is a new, distinct outbreak in the west. This sentence may need slightly updating accordingly.

Line 521 – schools in Sweden indeed remained open to under-16s but there was one school, in a relatively remote northern area, Skellefteå, that had to close as a result of high infection rates <https://www.sciencemag.org/news/2020/05/how-sweden-wasted-rare-opportunity-study-coronavirus-schools>

Line 613 – some specific tools for rapid assessment of COVID-19 mortality in data-poor settings have become available, for example from WHO/Vital Strategies, Rapid Mortality Surveillance and Response tools: <https://www.who.int/publications/i/item/revealing-the-toll-of-covid-19>

Decision letter (RSPB-2020-1405.R0)

07-Jul-2020

Dear Dr Thompson:

Your manuscript has now been peer reviewed and the reviews have been assessed by an Associate Editor. The reviewers' comments (not including confidential comments to the Editor) and the comments from the Associate Editor are included at the end of this email for your reference. As you will see, the reviewers and the Editors have raised some concerns with your manuscript and we would like to invite you to revise your manuscript to address them.

Research ethics:

Use of animals and field studies:

Please submit a copy of your revised paper within three weeks. If we do not hear from you within this time your manuscript will be rejected. If you are unable to meet this deadline please let us know as soon as possible, as we may be able to grant a short extension.

Best wishes,

Dr The Proceedings B Team

Associate Editor

Board Member: 1

Comments to Author:

Thank you for submitting your interesting manuscript for consideration as a PRSB Evidence Synthesis article. Your work has now been reviewed by 2 experts in the field, and I have read the manuscript myself. Collectively, I am pleased to say that there is a clear consensus that not only is the manuscript timely, interesting and written clearly, but importantly, within the lifetime of the pandemic, is likely to yield an identifiable and informed way forward. As such, I encourage you strongly to consider the comments below, and submit a revised manuscript at your earliest convenience. Especially, I echo the concerns of referee #2 relating to the potential impact of weather on transmission of Covid-19. There is of course, necessarily, a complexity and challenge in distinguishing between human social behaviour and the direct impacts of such climatic variables as temperature, humidity and so on, in relation to prevailing weather conditions. Some additional consideration and clarity here would be helpful. I fully appreciate the origins of this article, and the considerable value of providing a synthesis of collective opinion. However, I would like to draw your attention specifically to our requirements for publication of Evidence

Synthesis articles, which here, will require relatively little additional modification. Notwithstanding, in your response to referees, I would be grateful if you would include a brief account relating to Editorial comments, on how the manuscript has been modified in relation to my brief suggestions below. In particular, as you will have seen from the guidelines available for our Evidence Synthesis articles (<https://royalsocietypublishing.org/rspb/evidence-synthesis>), it is vital that the reader is able to assess the validity, robustness and objectivity of the evidence base presented. While I appreciate that much of the current understanding may come from models that are not fully validated, we need to ensure that there is a clear representation of particular published studies. I would therefore appreciate a brief account of how the literature base presented has been selected (in line with rough expectations for an evidence review, as detailed in the link above), and to what degree you have been able to secure the appropriate level of representation, objectivity and standardisation in studies cited. I appreciate this may simply be a brief qualitative description, but it is important in terms of transparency across our various similar such articles. Importantly also, when putting the final touches to the article, please ensure wherever possible, that where relevant, you have addressed some of the questions below, that characterises the Evidence Synthesis article type, though I fully recognise, that many questions will not only partially apply to your manuscript (in your case, the following are especially pertinent: 1,2,3,4,6,8,9,10:

1. Is the key policy-related question(s) articulated clearly?
2. Is there a clear justification in support of policy relevance?
3. Is the likely target audience identified clearly?
4. Does the search for literature utilise a comprehensive range of sources?
5. Does the synthesis article apply clearly documented inclusion criteria to all potentially relevant studies found during the search?
6. Is a clear methodology described to avoid bias?
7. Is your study objectively weighted according to methodological quality of cited literature?
8. Are knowledge gaps and priorities clearly identified?
9. Are outcomes/recommendations tangible in terms of likely impact?
10. Are all necessary supporting information available and accessible??

Thank you in advance for bringing this information together, and we look forward to receiving the revised manuscript in due course.

Reviewer(s)' Comments to Author:

Referee: 1

Comments to the Author(s)

This is an excellent review of the key issues regarding COVID-19 epidemic modelling at this stage of the pandemic. The paper focusses on the important and timely question of exit from lockdown. But I believe that it has wider relevance than that and will become a 'go to' reference concerning the use of epidemiological models during a public health emergency. None of the issues raised are new, but there is a need for re-stating the problems and solutions, if only because they are being re-visited during the current crisis.

I have a number of suggestions - all minor - that the authors might wish to consider.

L106. Personally, I'd delete the word "strong". There are several anomalies, not least the peaking of the epidemics in Sweden and Japan without strict 'lockdowns' (though social distancing was still surely a key factor). Ref. [1] reports an analysis that (to put it as diplomatically as possible) will surely not be the last word on the impact of lockdowns.

L157. I like the road map and it is highlighted again in the conclusions. But is the process really linear? I'd suggest there are feedbacks at every stage and particularly from the gathering of data to refinement of model structure and parameterisation. A good example for COVID-19 would be

the need to include care homes in any detailed model. That need emerged (sadly, rather belatedly) from observation; the early models missed it entirely.

L173. I'd insert "purely" here. In other epidemics (FMD 2001 being a good example) it was possible to estimate $R(t)$ directly from contact tracing data.

LL184-7. Here the concept of "instantaneous" R is raised. It's never fully explained and never mentioned again so seems a distraction.

L227. Interesting point about negative serial intervals. I'm unsure whether this actually occurs (indeed, I would be suspicious of the data if it did) but it does suggest a related complication: transmission from cases that are asymptomatic throughout.

LL242-3. Parentheses not required. That condition is not an add-on, it's crucial.

LL313-6. In general, the manuscript is extremely well written. But after several attempts I still couldn't make sense of this sentence.

Section 1.4. This section makes an important but slightly unsatisfying point about epidemic tails. The empirical evidence for COVID-19 epidemic tails is already plain to see. But the manuscript is less explicit about what models say about mechanisms. Presumably one answer is spatial (and presumably other) kinds of heterogeneity. Are there others? Also, what happens to $R(t)$ during a tail? Empirically it will rise to be close to 1, but that may not reflect any change in the underlying dynamics - a looming point of confusion as case numbers fall in the UK and elsewhere.

LL542-544. The ordering of these types of transmission suggests the wrong prioritisation. There is good evidence now from the world and from the UK, that teacher-teacher and teacher-child transmissions are the most important in a school. Child-teacher transmissions are extremely rare: none from any published study so far that I am aware of, and just one instance from PHE data on school "outbreaks".

Section 3, Data Needs. This section jars slightly because of an apparent shifting of attention to LMICs, which are not emphasized in previous sections. I noted this because the nature and quality of the epidemic data available in the UK is a huge concern, even now. Lack of clarity about sampling frame, sometimes even lack of denominator data, has made epidemiological analysis more difficult.

I suppose ref. 98 is fairly described as a starting point but it is too incomplete and error-strewn to be used uncritically in any formal analysis.

Referee: 2

Comments to the Author(s)

This is an interesting and insightful discussion of the complexities of modelling COVID-19 exit strategies, with a global perspective.

One aspect that does not appear to have been considered is the role of weather conditions in COVID transmission as part of developing modelling - while there seems to be consensus that COVID-19 transmission is not likely to be halted by any particular set of weather conditions, there are a number of published studies suggesting that factors such as temperature, humidity, precipitation, solar radiation, etc. have effects of significant sizes on transmissibility. It is hard to separate the direct effects of weather on the virus in outdoor environments from the modulating effect of weather on human behaviour in relation to compliance with lockdown conditions and balance of time spent in indoor and outdoor environments. A number of studies have noted that weather conditions in the major foci of Wuhan, Daegu, Qom and Lombardy in January/February 2020 were closely similar. It is reasonable to consider that prevailing weather conditions in a

particular place and time might be able to tip $R(t)$ to one side or the other of one, and thus be the basis for tweaking other NPI components accordingly. One good thing is that global gridded weather data, for example the ERA5-T model, are available in high resolution globally and are in the public domain.

Some minor points:

Line 362 – my understanding is that the long-running Ebola outbreak in eastern DR Congo has now been declared over, although there is a new, distinct outbreak in the west. This sentence may need slightly updating accordingly.

Line 521 – schools in Sweden indeed remained open to under-16s but there was one school, in a relatively remote northern area, Skellefteå, that had to close as a result of high infection rates <https://www.sciencemag.org/news/2020/05/how-sweden-wasted-rare-opportunity-study-coronavirus-schools>

Line 613 – some specific tools for rapid assessment of COVID-19 mortality in data-poor settings have become available, for example from WHO/Vital Strategies, Rapid Mortality Surveillance and Response tools: <https://www.who.int/publications/i/item/revealing-the-toll-of-covid-19>

Author's Response to Decision Letter for (RSPB-2020-1405.R0)

See Appendix A.

Decision letter (RSPB-2020-1405.R1)

21-Jul-2020

Dear Dr Thompson

I am pleased to inform you that your manuscript entitled "Key Questions for Modelling COVID-19 Exit Strategies" has been accepted for publication in Proceedings B.

Open Access

Your article has been estimated as being 16 pages long. Our Production Office will be able to confirm the exact length at proof stage.

Paper charges

Sincerely,

Gary Carvalho

Comments to Author:

Thank you for your constructive responses to issues raised on the above manuscript. I very much appreciate both the full detail of your responses, justification, and a clear cross-referencing to changes made in the manuscript. I am now happy to recommend acceptance of your manuscript, and very much look forward to seeing this timely and important document published as soon as possible. Thank you once again for your interest in publishing in PRSB, and in contributing to our Evidence Synthesis articles.

Appendix A

Robin N. Thompson
Christ Church
University of Oxford
St Aldates, Oxford
OX1 1DP, UK

21 July 2020

Corresponding author tel: +44 7526 330 080

Corresponding author email: robin.thompson@chch.ox.ac.uk

Dear Editor,

Key Questions for Modelling COVID-19 Exit Strategies

Robin N. Thompson, T. Déirdre Hollingsworth, Valerie Isham, Daniel Arribas-Bel, Ben Ashby, Tom Britton, Peter Challoner, Lauren H. K. Chappell, Hannah Clapham, Nik J. Cunniffe, A. Philip Dawid, Christl A. Donnelly, Rosalind Eggo, Sebastian Funk, Nigel Gilbert, Julia R. Gog, Paul Glendinning, William S. Hart, Hans Heesterbeek, Thomas House, Matt Keeling, Istvan Z. Kiss, Mirjam Kretzschmar, Alun L. Lloyd, Emma S. McBryde, James M. McCaw, Joel C. Miller, Trevelyan J. McKinley, Martina Morris, Philip D. O'Neill, Carl A. B. Pearson, Kris V. Parag, Lorenzo Pellis, Juliet R. C. Pulliam, Joshua V. Ross, Michael J. Tildesley, Gianpaolo Scalia Tomba, Bernard W. Silverman, Claudio J. Struchiner, Pieter Trapman, Cerian R. Webb, Denis Mollison, Olivier Restif

Thank you for arranging for such useful reviews and for encouraging us to revise the above-named manuscript. We are grateful to the associate editor and referees for their helpful comments and careful reading of the submission.

We have now made all the changes suggested by the associate editor and referees. Please find below a response in which we explain and itemise these changes. As requested by the associate editor and referee 2, our most substantive change is to address the important question of the impact of weather on COVID-19 directly (lines 1023-1040 of the “track changes” version of the revised manuscript).

Thank you for handling our manuscript, and we hope that you will now find it appropriate for publication as an Evidence Synthesis article in *Proceedings of the Royal Society B*.

Yours faithfully, on behalf of all authors,

Robin Thompson

Please note in the below that whenever we refer to line numbers in the revised manuscript, these refer to the version of the manuscript with “track changes”. We have uploaded two versions of the revised manuscript – one with, and one without, track changes.

Associate Editor

Thank you for submitting your interesting manuscript for consideration as a PRSB Evidence Synthesis article. Your work has now been reviewed by 2 experts in the field, and I have read the manuscript myself. Collectively, I am pleased to say that there is a clear consensus that not only is the manuscript timely, interesting and written clearly, but importantly, within the lifetime of the pandemic, is likely to yield an identifiable and informed way forward. As such, I encourage you strongly to consider the comments below, and submit a revised manuscript at your earliest convenience.

***Response:** Thank you very much for these positive comments. We are pleased that the current importance of our manuscript has been recognised, and that we are able to provide a clear and informed way forwards in the pandemic. As suggested, we have addressed the comments below fully in our revised submission.*

Especially, I echo the concerns of referee #2 relating to the potential impact of weather on transmission of Covid-19. There is of course, necessarily, a complexity and challenge in distinguishing between human social behaviour and the direct impacts of such climatic variables as temperature, humidity and so on, in relation to prevailing weather conditions. Some additional consideration and clarity here would be helpful.

***Response:** We agree that the weather could be an important factor in the COVID-19 pandemic, potentially affecting transmission of SARS-CoV-2 in different countries. As you suggest, unpicking the effects of weather from the many other factors that impact transmission is complex – and this is a key challenge going forwards.*

***Action:** We have added a new paragraph to Section 3.2 in which we outline the importance of differences between countries on SARS-CoV-2 transmission. There are many different factors that could be responsible for different patterns of case numbers in different locations. These include the weather (with recent evidence suggesting potential correlations between weather variables and SARS-CoV-2 transmissibility), but also different population structures between countries and the stage of the epidemic when cases were first detected. We now have summarised these issues (lines 999-1040 of the “track changes” version of the revised manuscript).*

I fully appreciate the origins of this article, and the considerable value of providing a synthesis of collective opinion. However, I would like to draw your attention specifically to our requirements for publication of Evidence Synthesis articles, which here, will require relatively little additional modification. Notwithstanding, in your response to

referees, I would be grateful if you would include a brief account relating to Editorial comments, on how the manuscript has been modified in relation to my brief suggestions below. In particular, as you will have seen from the guidelines available for our Evidence Synthesis articles (<https://royalsocietypublishing.org/rspb/evidence-synthesis>), it is vital that the reader is able to assess the validity, robustness and objectivity of the evidence base presented. While I appreciate that much of the current understanding may come from models that are not fully validated, we need to ensure that there is a clear representation of particular published studies. I would therefore appreciate a brief account of how the literature base presented has been selected (in line with rough expectations for an evidence review, as detailed in the link above), and to what degree you have been able to secure the appropriate level of representation, objectivity and standardisation in studies cited. I appreciate this may simply be a brief qualitative description, but it is important in terms of transparency across our various similar such articles. Importantly also, when putting the final touches to the article, please ensure wherever possible, that where relevant, you have addressed some of the questions below, that characterises the Evidence Synthesis article type, though I fully recognise, that many questions will not only partially apply to your manuscript (in your case, the following are especially pertinent: 1,2,3,4,6,8,9,10:

1. Is the key policy-related question(s) articulated clearly?
2. Is there a clear justification in support of policy relevance?
3. Is the likely target audience identified clearly?
4. Does the search for literature utilise a comprehensive range of sources?
5. Does the synthesis article apply clearly documented inclusion criteria to all potentially relevant studies found during the search?
6. Is a clear methodology described to avoid bias?
7. Is your study objectively weighted according to methodological quality of cited literature?
8. Are knowledge gaps and priorities clearly identified?
9. Are outcomes/recommendations tangible in terms of likely impact?
10. Are all necessary supporting information available and accessible?

Thank you in advance for bringing this information together, and we look forward to receiving the revised manuscript in due course.

Response and Action: *Thank you for directing us to the guidelines for Evidence Synthesis articles, which we have considered in detail when putting together the revised manuscript. We appreciate the need to ensure that there is a clear representation of particular published studies when collating and summarising studies related to a specific policy-relevant question. As requested, we have included in the revised manuscript a qualitative description as to precisely how the evidence that we present has been selected (lines 174-183). We have also ensured that the questions that you have picked out as being particularly pertinent for our manuscript have been addressed directly, as described below.*

Question 1: The overriding policy-related question addressed in this article is “What are the key questions that, if answered, will allow for more accurate predictions of the effects of different exit strategies?” This question is stated clearly in the abstract, and then answered directly in the main text of the manuscript (with each subsection corresponding to one of the questions identified).

Question 2: We provide a roadmap for future research. The justification for this in terms of policy relevance is given in lines 213-216 of the revised manuscript: “If this roadmap can be followed, it will be possible for policy-makers to predict the effects of different potential exit strategies with more precision. This is of clear benefit to global health, allowing exit strategies to be chosen that allow interventions to be relaxed while limiting the risk of transmission.”

Question 3: Our Evidence Synthesis manuscript is aimed at a diverse scientific audience, since the roadmap of research that we have proposed requires a global collaborative effort from the entire scientific community. We have emphasised this throughout (e.g. “The roadmap requires a global collaborative effort from the scientific community and policy-makers” – lines 109-110; “[we] provide a roadmap for modellers and other scientists” – lines 171; “These are international challenges that require an international collaborative response from diverse scientific communities, which we hope that this article will stimulate” – lines 1221-1227).

Question 4: In this manuscript, we use a wide range of sources, and have explained exactly how these were selected in the revised manuscript (lines 174-180).

Question 6: The approaches that we took to avoid bias are described in lines 180-183 of the revised manuscript.

Question 8: We outlined the key knowledge gaps and priorities clearly in our manuscript, with the main knowledge gaps being summarised in distinct subsections corresponding to different key questions. In the middle of a pandemic, there are many competing priorities all of which are very important. The roadmap for future research lays out a clear way forwards for researchers, and we contend that the different aspects of the roadmap can be carried out concurrently with useful feedbacks occurring between the different areas.

Question 9: As identified by the reviewers, the recommendations in this manuscript are likely to have significant impacts on the research that is being carried out. Since countries worldwide are informing their policy responses to COVID-19 with results from epidemiological models, improving the models will have clear positive impacts and will lead to a clearer understanding of SARS-CoV-2 transmission and optimal exit strategies.

Question 10: The only supporting information included with our manuscript is Figure S1. When we cite reports that are not published on standard preprint servers or in peer reviewed journals, we provide URLs so that the relevant material can be accessed by the reader.

Referee 1

This is an excellent review of the key issues regarding COVID-19 epidemic modelling at this stage of the pandemic. The paper focusses on the important and timely question of exit from lockdown. But I believe that it has wider relevance than that and will become a 'go to' reference concerning the use of epidemiological models during a public health emergency. None of the issues raised are new, but there is a need for re-stating the problems and solutions, if only because they are being re-visited during the current crisis.

I have a number of suggestions - all minor - that the authors might wish to consider.

Response: *Thank you very much for these positive comments. We have made all the changes suggested in the revised manuscript.*

L106. Personally, I'd delete the word "strong". There are several anomalies, not least the peaking of the epidemics in Sweden and Japan without strict 'lockdowns' (though social distancing was still surely a key factor). Ref. [1] reports an analysis that (to put it as diplomatically as possible) will surely not be the last word on the impact of lockdowns.

Response and Action: *We have made the change suggested, as well as noting the peaking of the outbreaks in Sweden and Japan without strict lockdowns being in place.*

L157. I like the road map and it is highlighted again in the conclusions. But is the process really linear? I'd suggest there are feedbacks at every stage and particularly from the gathering of data to refinement of model structure and parameterisation. A good example for COVID-19 would be the need to include care homes in any detailed model. That need emerged (sadly, rather belatedly) from observation; the early models missed it entirely.

Response: *The reviewer is absolutely correct that the proposed roadmap is not necessarily a linear process, and that there are feedbacks between the different aspects of the proposed research.*

Action: *We have now stated more explicitly that the roadmap is not a linear process (lines 199-213 and 225-226) and changed the arrows in Fig 1 to reflect this.*

L173. I'd insert "purely" here. In other epidemics (FMD 2001 being a good example) it was possible to estimate $R(t)$ directly from contact tracing data.

Response and Action: *Thank you for pointing out that this was not sufficiently clear. We have now differentiated estimation of $R(t)$ using case notification data from other possible approaches by noting the example of estimating $R(t)$ using contact tracing data.*

LL184-7. Here the concept of "instantaneous" R is raised. It's never fully explained and never mentioned again so seems a distraction.

Response and Action: *We agree that it was not necessary to introduce this concept and have now deleted this sentence to simplify the revised manuscript.*

L227. Interesting point about negative serial intervals. I'm unsure whether this actually occurs (indeed, I would be suspicious of the data if it did) but it does suggest a related complication: transmission from cases that are asymptomatic throughout.

Response and Action: *In fact, recent publications suggest that negative serial intervals can (and indeed have) occurred in some SARS-CoV-2 transmission chains. We have cited these relevant publications in the revised manuscript (notably, reference 42), as well as mentioning that asymptomatic infected individuals (i.e. those who never develop symptoms) make estimating $R(t)$ more challenging.*

LL242-3. Parentheses not required. That condition is not an add-on, it's crucial.

Response and Action: *We have removed the parentheses as suggested.*

LL313-6. In general, the manuscript is extremely well written. But after several attempts I still couldn't make sense of this sentence.

Response and Action: *Thank you for pointing out that this was unclear. We have now rephrased this sentence.*

Section 1.4. This section makes an important but slightly unsatisfying point about epidemic tails. The empirical evidence for COVID-19 epidemic tails is already plain to see. But the manuscript is less explicit about what models say about mechanisms. Presumably one answer is spatial (and presumably other) kinds of heterogeneity. Are there others? Also, what happens to $R(t)$ during a tail? Empirically it will rise to be close to 1, but that may not reflect any change in the underlying dynamics - a looming point of confusion as case numbers fall in the UK and elsewhere.

Response: *The reviewer is absolutely correct that different types of heterogeneity, including spatial heterogeneity (in weather patterns, access to treatment, and a range of other factors) can drive long epidemic tails. At a local level, virus reimportation from elsewhere can extend outbreaks that appear to be over (e.g. the reimportation of SARS-CoV-2 to New Zealand from the UK in June).*

As the reviewer states, the challenge of estimating $R(t)$ when case numbers are low is a difficult one to address. Methods for estimating reproduction numbers that are based on the assumption that $R(t)$ is fixed within pre-specified time windows can be used to estimate $R(t)$ more robustly when there are few cases by making the assumption of a long window length, thereby preventing reversion to the prior for $R(t)$ – however, estimates are then less sensitive to temporal variations in transmissibility. Alternative approaches include estimating the transmission potential from other sources of data, such as contact surveys.

Action: *In the revised manuscript, we have discussed potential mechanisms that can lead to long outbreak tails in more detail (lines 531-536). We have now also described the issues of estimating $R(t)$ when case numbers are low – including in the tails of epidemics – in an earlier section of the manuscript, noting the relevance in the context of Section 1.4 (lines 317-324).*

LL542-544. The ordering of these types of transmission suggests the wrong prioritisation. There is good evidence now from the world and from the UK, that teacher-teacher and teacher-child transmissions are the most important in a school. Child-teacher transmissions are extremely rare: none from any published study so far that I am aware of, and just one instance from PHE data on school "outbreaks".

Response and Action: *We agree and have now changed the ordering here to reflect this.*

Section 3, Data Needs. This section jars slightly because of an apparent shifting of attention to LMICs, which are not emphasized in previous sections. I noted this because the nature and quality of the epidemic data available in the UK is a huge concern, even now. Lack of clarity about sampling frame, sometimes even lack of denominator data, has made epidemiological analysis more difficult.

Response and Action: *Thank you for pointing this out. We believe that challenges in data limited settings are very important going forwards, and therefore deserving of an entire subsection of our manuscript. However, we agree that this subsection did not follow naturally from previous sections of the manuscript, and so we have added text at the beginning of this subsection to ease this transition in focus.*

I suppose ref. 98 is fairly described as a starting point but it is too incomplete and error-strewn to be used uncritically in any formal analysis.

Response and Action: *Given the issues raised by the reviewer, we have removed this, to avoid appearing to endorse this tracker as a starting point for cross country comparisons.*

Thank you again for your very helpful comments that have enabled us to improve the manuscript.

Referee 2

This is an interesting and insightful discussion of the complexities of modelling COVID-19 exit strategies, with a global perspective.

Response: *Thank you for this positive comment – we are glad that you found our manuscript interesting and that we were able to provide useful insights.*

One aspect that does not appear to have been considered is the role of weather conditions in COVID transmission as part of developing modelling - while there seems to be consensus that COVID-19 transmission is not likely to be halted by any particular set of weather conditions, there are a number of published studies suggesting that factors such as temperature, humidity, precipitation, solar radiation, etc. have effects of significant sizes on transmissibility. It is hard to separate the direct effects of weather on the virus in outdoor environments from the modulating effect of weather on human behaviour in relation to compliance with lockdown conditions and balance of time spent in indoor and outdoor environments. A number of studies have noted that weather conditions in the major foci of Wuhan, Daegu, Qom and Lombardy in January/February 2020 were closely similar. It is reasonable to consider that prevailing weather conditions in a particular place and time might be able to tip $R(t)$ to one side or the other of one, and thus be the basis for tweaking other NPI components accordingly. One good thing is that global gridded weather data, for example the ERA5-T model, are available in high resolution globally and are in the public domain.

Response and Action: *Thank you for this excellent suggestion. We agree that the impact of weather on the COVID-19 pandemic is an important consideration in the context of exit strategies. Differences in the weather between countries are likely to lead to different case patterns, although – as the reviewer states – it is challenging to separate the effects of weather from a range of other factors. We have now added a new paragraph to the manuscript in which we discuss these issues in more detail (lines 999-1040 of the revised manuscript).*

Some minor points:

Line 362 – my understanding is that the long-running Ebola outbreak in eastern DR Congo has now been declared over, although there is a new, distinct outbreak in the west. This sentence may need slightly updating accordingly.

Response and Action: *We have now amended this, and updated the citation accordingly.*

Line 521 – schools in Sweden indeed remained open to under-16s but there was one school, in a relatively remote northern area, Skellefteå, that had to close as a result of high infection rates <https://www.sciencemag.org/news/2020/05/how-sweden-wasted->

rare-opportunity-study-coronavirus-schools

Line 613 – some specific tools for rapid assessment of COVID-19 mortality in data-poor settings have become available, for example from WHO/Vital Strategies, Rapid Mortality Surveillance and Response tools: <https://www.who.int/publications/i/item/revealing-the-toll-of-covid-19>

Response and Action: *Thank you for these useful points, which we have noted in the revised manuscript.*

Thank you again for your very helpful comments, particularly the suggestion to address directly the relationship between weather and COVID-19. Your suggestions have enabled us to improve the manuscript.